



# Assessing the impact of the Kuroshio Current on vertical cloud structure using CloudSat data

Akira Yamauchi[1*], Kazuaki Kawamoto[1], Atsuyoshi Manda[1, 2], and Jiming Li[3]

[1]School of Fisheries Science and Environmental Sciences, Nagasaki University, Nagasaki, Japan
[2]Earth and Environmental Sciences Division, Graduate School of Bioresources, Mie University, Mie, Japan
[3]Key Laboratory for Semi-Arid Climate Change of the Ministry of Education, College of Atmospheric Sciences, Lanzhou University, Lanzhou, China

*Correspondence to: A. Yamauchi (akira19890620@gmail.com)

**Abstract.** This study analysed CloudSat satellite data to determine how the warm ocean Kuroshio Current affects the vertical structure of clouds. Rainfall intensity around the middle troposphere (6 km in height) over the Kuroshio was greater than that over surrounding areas. The drizzle clouds over the Kuroshio have a higher frequency of occurrence of

geometrically thin (0.5–3 km) clouds and thicker (7–10 km) clouds compared to those around the Kuroshio. Moreover, the frequency of occurrence of precipitating clouds with a geometric thickness of 7 to 10 km increased over the Kuroshio. Stronger updraft over the Kuroshio maintains large droplets higher in the upper part of the cloud layer, and the maximum radar reflectivity within a cloud layer in non-precipitating and drizzle clouds over the Kuroshio is higher than that around the Kuroshio.

## 1 Introduction

Clouds are recognized as one of the primary sources of uncertainty in understanding and predicting global climate change (e.g., Stephens, 2005; Dufresne and Bony, 2008). As a key component of the climate system, clouds have a significant influence on hydrological cycles and energy budgets. Cloud effects are strongly regulated by their microphysical (particle

size, number concentration, and mass density of water or ice particulates), and macro physical (temporal frequency, height,



geometrical thickness, and rainfall intensity) structure; thus, an investigation of the internal structure of clouds provides insight into various atmospheric phenomena. For example, Kawamoto and Hayasaka (2008) reported that the surface radiative properties were crucially regulated by cloud optical thickness and cloud cover. The previous generation circulation models (GCMs) underestimated (overestimated) the cloud fraction (the radiative effect) in tropical low-level clouds (Nam et

al., 2012). Moreover, Suzuki et al. (2015) reported that warm cloud auto-conversion process in the GCMs was too rapid compared to that derived from satellite observation. It is necessary to understand the vertical structure of clouds to reduce the cloud bias in the GCMs.

The advent of active remote-sensing data has provided a more detailed picture of some cloud effects. CloudSat is a part of a satellite constellation of passive and active sensors called the A-train, which crosses the equator within a few minutes of

one another at about 1:30 pm (13:30) local time from a 705 km altitude orbit (Stephens et al., 2002). CloudSat carries a cloud profiling radar (CPR) that operates at 94 GHz. Its footprint is approximately 1.7 km along-track by 1.3 km across-track. One granule consists of approximately 36,383 profiles, and one profile has 125 vertical bins, each of which is 240 m thick. The CloudSat CPR is sensitive to cloud particle size, which ranges from drizzle to precipitation (Haynes et al., 2009). Huang et al. (2016) reported that liquid-phase cloud properties (cloud top temperature, effective radius, cloud optical thickness, and

liquid water path) of the Southern Ocean strongly correlate with the sea surface temperature (SST) using A-train satellite observations. In this study, we clarified how the vertical structure (rainfall intensity, cloud geometrical thickness, and maximum radar reflectivity position) was affected by strong upward wind flow over the Kuroshio Current. The Kuroshio, a western boundary current of the North Pacific Subtropical Gyre, flows northward along the east coast of Taiwan into the East China Sea (ECS). This current carries warm water from the tropics and forms a warm tongue in the SST field in the

ECS. The recent studies indicate the influence of the Kuroshio SST on the atmosphere of the ECS is as strong as other western boundary currents such as the Gulf Stream and Agulhas Current (e.g., Xie et al., 2004; Small et al., 2008). Evaporation from the sea surface in the Kuroshio region enhances the formation of convectively unstable air masses, which can produce deep convection (Kuwano-Yoshida 2013; Tsuguti and Kato 2014; Manda et al., 2014; Kunoki et al., 2015; Sato et al., 2016). The Kuroshio also impacts the surface wind convergence, which in turn, helps form the deep convective clouds

(Xu et al., 2011; Sasaki et al., 2012; Miyama et al., 2012). Recent studies reported the impact of the Kuroshio on the cloud



vertical structure. High upward flow over the Kuroshio affected the vertical structure and microphysics of low altitude non-precipitating water clouds (Koike et al., 2012, 2016). Liu et al. (2016) reported the low cloud response to the Kuroshio from winter to spring. However, our understanding of the impact of the Kuroshio on the vertical structure and hydrometeor over the Kuroshio is still limited. In this study, we investigate how the sea surface temperature changes around the Kuroshio

affect the vertical structure and hydrometeor of clouds during early summer.

This paper is organized as follows. In section 2, we showed the data, target region, and method. In Section 3.1, we described the influence of the Kuroshio on atmospheric conditions. In Section 3.2, we presented the relationship between radar reflectivity and altitude. The relationship between geometrically thick clouds and rainfall intensity was discussed in the Section 3.3. Finally, in Section 3.4, we described the maximum radar reflectivity position inside clouds. In section 4, we

summarized our overall conclusions.

**2 Data and Methods**

In this study, we used the CloudSat product 2B-GEOPROF 240 m resolution vertical distribution of the cloud mask, radar reflectivity (Mace et al., 2007; Marchand et al., 2008), altitude, and temperature profiles from the European Center for

Medium-Range Weather Forecasts (ECMWF-AUX) (Partain, 2007), precipitating liquid and ice water content from 2C-RAIN-PROFILE (Mitrescu et al., 2010; Lebsock et al., 2011), heating rates form 2B-FLXHR (L'Ecuyer et al. 2008), and the Japan Meteorological Agency's Mesoscale Model (MSM, $0.125 \times 0.1°$, 16 levels) for the period between May 15 and June 15 from 2007 to 2010. This analysis period was selected because the SST gradient in the ECS is strongest during this time of the year, providing the greatest SST contrast between the Kuroshio and surrounding areas. The target region was a section of

the ECS (25–34°N, 120–131°E, the yellow square area shown in Figure 1a). Hereafter, areas with an SST warmer than 24°C are referred to as "ON Kuroshio," and those with temperatures below 24°C are referred to as "OFF Kuroshio". We defined the representative value of the SST in the Kuroshio front as that with the steepest SST gradient. The results did not significantly change when using 23°C or 25°C as thresholds.

We defined 'clouds' as layers with a cloud mask value between 30 and 40. An increase in the cloud mask indicated the

reduced probability of false detection. Cloud mask values between 30 and 40 represented high-confidence detections (<2%



false detections). In this study, we defined thick clouds with a geometric thickness >6 km as convective clouds. For

classification purposes, we adopted three precipitation categories: non-precipitating, drizzle, and precipitating, corresponding

to radar reflectivity <–15dBZ, –15 < radar reflectivity < 0 dBZ, and radar reflectivity >0 dBZ, respectively (e.g., L'Ecuyer et

al., 2009; Suzuki et al., 2011). Lower-tropospheric stability (LTS) is defined as the difference in potential temperatures

between 700 hPa and surface (Klein and Hartmann, 1993) using ECMWF-AUX.

### 3 Results

### 3.1 Relationship among upward velocity, rainfall intensity, and frequency of cloud occurrence

Before examining the influence of the Kuroshio on cloud structure, we first describe the influence of the Kuroshio on

atmospheric conditions. Figure 1a shows the average wind convergence (divergence) from the MSM at 1000 hPa for the

target period. Figure 1d presents the mean skin temperature from ECMWF-AUX. Wind convergence was found at warmer

flank around the Kuroshio front. Figure 1b shows the mean upward velocity of the MSM at 400 hPa. Strong updrafts

occurred in the ECS during the target period and were strongest at 400 hPa. This observation was consistent with a previous

report (Sasaki et al., 2012). Figure 1c shows the mean relative humidity of the MSM at 400 hPa. Relative humidity around

the Kuroshio was higher than the domain of ECS, with the exception of the Kuroshio. Figure 1e presents the mean LTS

using ECMWF-AUX. LTS over the Kuroshio was significantly lower compared to around Kuroshio. Figure 1f shows the

mean cloud fraction of the CloudSat cloud mask (from 2B-GEOPROF). Cloud fraction was mostly high over the Kuroshio,

although it was high throughout the ECS. In addition to this result, the data presented in the thick dotted square (28–31.5°N,

120–131°E) of Figure 1a were divided into eleven zonal vertical sections: the average wind convergence (Fig. 2a), the mean

upward velocity (Fig. 2b), the mean relative humidity (Fig. 2c), the cloud fraction (Fig. 2d), the mean cloud liquid water

content (Fig. 2e), the mean precipitating liquid water content (Fig. 2f), the mean precipitating ice water content (Fig. 2g), the

mean total water content (Fig. 2h), the mean shortwave cloud radiative heating rate (Fig. 2i), the mean longwave cloud

radiative heating rate (Fig. 2j), and the mean net cloud radiative heating rate (Fig. 2k). Wind convergence (Fig. 2a) was

found from the lower troposphere over the Kuroshio (126–130°E), and the mean upward velocity (Fig. 2b) was strongest

around 400 hPa over the Kuroshio and became weak or negative at lower longitudes. The relative humidity (Fig. 2c) in the





lower to upper troposphere over the Kuroshio was higher than the west side of the Kuroshio (less than 126E°). The cloud

fraction (Fig. 2d) in the lower to upper troposphere was high in the ECS as a whole, and there was no clear difference

between over the Kuroshio and around the Kuroshio. Note that Figure 2d is a frequency of the presence of clouds, and is not

related to the rainfall intensity. We observed that the precipitating liquid and ice water content (PLWC and PIWC, Fig. 2f, g)

were high from the lower to the middle troposphere (0.5–6 km) over the Eastside area of the SST front of the Kuroshio

(126°E). However, the cloud liquid water content (CWC) did not show a clear difference between over the Kuroshio and

around the Kuroshio. The Kuroshio affected the lower to middle-level PLWC and PIWC inside clouds more than the CWC

(Fig. 2e, g). The total water content (TWC: CWC+PLWC+PIWC) corresponded to LTS, which peaked around 124.5E°. In

addition, TWC increased with decreasing LTS. The stronger upward motion and lower LTS over the Kuroshio could change

the PLWC, PIWC, and TWC.

     Finally, we considered the influence of clouds over the Kuroshio on atmospheric radiative processes. Shortwave

radiative absorption and longwave radiative emission act to heat and cool the atmosphere, respectively (Fig. 2i, j). Radiative

heating occurred due to shortwave radiation inside the cloud, while radiative cooling occurred due to longwave radiation

around the cloud top (Yoshida et al., 2004). Shortwave cloud radiative heating and longwave cloud radiative cooling were

both strong at the higher altitudes over the Kuroshio (127–128°E, Fig. 2i, j). Strong shortwave cloud radiative heating

occurred in the middle to upper layer (7-15 km, Fig. 2i), while strong longwave radiative cooling occurred in the upper layer

(14-15 km, Fig. 2j) over the Kuroshio. In summary, net radiative heating over the Kuroshio occurred at a height of around 14

km (Fig. 2k). As previously described, TWC increased over the Kuroshio. Shortwave radiative absorption inside the cloud

was enhanced by this change, with Stephens (1978) suggesting that it increased as the total amount of cloud water increased.

From this analysis, it was concluded that clouds over the Kuroshio had a role in heating the atmosphere at higher altitudes.

### 3.2 Two-dimensional Probability Density Function and Contoured Frequency by Altitude Diagrams

In this section, we discuss the effects of altitude on rainfall intensity over the Kuroshio. Figure 3a to 3c shows the two-

dimensional probability density function (PDF) of radar reflectivity, and Figure 3d to 3f shows the contoured frequency by

altitude diagrams (CFADs, Yuter and Houze 1995) that represent the relationship between radar reflectivity and cloud



altitude. Only the lowest cloud in each CloudSat profile was included in the analysis, and the target region was a section of

the ECS (25–34°N, 120–131°E, the yellow square area shown in Fig. 1a). The peak frequency of occurrence lay around 0 to

15 dBZ and 1 to 6 km (Fig. 3a, b). These results show that clouds with high rainfall intensity at lower altitudes (1–6 km) are

common in the target region. The radar reflectivity was weak (<0 dBZ) both ON and OFF Kuroshio in the lower layer, but

gradually became stronger (>0 dBz) closer to the middle layer (~6 km) as the altitude increased (Fig. 3d, e). Above 6 km, the

radar reflectivity weakened as the altitude increased. Although the ON and OFF Kuroshio showed similar tendencies, some

differences were evident (Figure 3c, f). High radar reflectivity was more common in the OFF Kuroshio up to ~6 km;

however, at higher altitudes, high reflectivity was more common for the ON Kuroshio (Fig. 3c, f). These differences were

statistically significant at a 95 % confidence level using Student's t-test. These results indicate that the altitude of the peak

rainfall intensity was different between ON and OFF Kuroshio, and the Kuroshio affected the clouds in the middle

troposphere. The radar reflectivity of the middle clouds (around 6 km) was higher ON Kuroshio than OFF Kuroshio. At mid-

altitudes, the average precipitation intensity was stronger for ON Kuroshio than for OFF Kuroshio.

### 3.3 Relationship between cloud geometric thickness and radar reflectivity

In this section, we discuss the relationship between cloud geometrical thickness and rainfall properties inside the cloud. The

difference in the frequency (ON Kuroshio – OFF Kuroshio) of the geometric thicknesses is shown in Figure 4 according to

three categories of maximum radar reflectivity (MaxZe) for (a) non-precipitating (MaxZe <–15 dBZ), (b) drizzle (-15 <

MaxZe < 0 dBZ), and (c) precipitating (MaxZe > 0 dBZ) clouds. Here, the MaxZe refers to the maximum radar reflectivity

value in the cloud. These thresholds for the occurrence of precipitation were taken from L'Ecuyer et al. (2009). Only the

lowest cloud in each CloudSat profile was included in the analysis, and the target region was a section of the ECS (25–34°N,

120–131°E, the yellow square area shown in Figure 1a). The non-precipitating clouds included many geometrically thin

clouds both ON and OFF Kuroshio, and the drizzle and the precipitating clouds included many geometrically thick clouds.

The fraction of occurrence of non-precipitating clouds is not much different between ON and OFF Kuroshio (Fig. 4a). The

drizzle clouds over ON Kuroshio have a higher chance of being geometrically thin (0.5–3 km) and thicker (7–10 km)





compared to those over OFF Kuroshio (Fig. 4b). The drizzle clouds over ON Kuroshio exist in geometrically thin and thick

cases. Rainfall intensity of thin clouds over the ON Kuroshio could be enhanced by Kuroshio. In addition, the frequency of

occurrence of the precipitating clouds with a geometric thickness of 7 to 10 km significantly increased ON Kuroshio.

Precipitating clouds reveal a distinct difference between the ON and OFF Kuroshio regions, with the Kuroshio considered to

have a more significant impact on drizzle and precipitating clouds than on non-precipitating clouds. Consequently, these

geometrically thick clouds with precipitation inside the cloud could be caused by stronger updrafts over the Kuroshio;

additionally, these clouds corresponded to the convective rain clouds reported by Miyama et al. (2012).

**3.4 Relationship between maximum radar reflectivity position and radar reflectivity.**

Figure 5a to 5c shows the relationship between the fraction of maximum radar reflectivity (MaxZe) position and altitude

according to the three categories (non-precipitating, drizzle, and precipitating) of MaxZe. Only the lowest cloud in each

CloudSat profile was included in the analysis, and the target region was a section of the ECS (25–34°N, 120–131°E, the

yellow square area shown in Figure 1a). The fraction of the MaxZe position in non-precipitating clouds reaches a peak

around high altitude (10–12 km). As MaxZe increases, the altitude of MaxZe position decreases. In the case of non-

precipitating and drizzle clouds, the fraction of MaxZe in the lower layer (2 to 5 km) over the Kuroshio slightly increases

compared to OFF Kuroshio. These results show that the high relative humidity and the dynamic vertical motion over the

Kuroshio could influence drizzle cloud from the lower troposphere.

    To determine the relative position of MaxZe in clouds, vertical normalization was undertaken such that the cloud top is 0

and the cloud base is 1. Each 240 m thick cloud layer was assigned to 10 sublayers between the cloud top (0) and the cloud

base (1). In this analysis, we used clouds with more than three layers. For example, for the three-layer cloud, the first, second,

and third layers were assigned 0, 0.5, and 1, respectively. For the five-layer cloud, the first, second, third, fourth, and fifth

layers were assigned 0, 0.25, 0.5, 0.75, and 1, respectively. For example, when the second layer in the fifth layer cloud

shows the maximum radar reflectivity, the vertical normalization was rounded off to 0.3.  Figure 5d to 5f show the fraction

of the MaxZe position for the scaled cloud layer according to the three categories of MaxZe. The horizontal axis is the

fraction of MaxZe position, and the vertical axis is the relative position from the cloud top, taking 0 and 1 at the cloud top

and the cloud base, respectively. The radar reflectivity almost reaches a maximum near the center of the cloud (Fig. 5d, f).




However, characteristics of non-precipitating and drizzle clouds are slightly different between ON and OFF Kuroshio. The MaxZe position in non-precipitating and drizzle clouds over ON Kuroshio is higher than OFF Kuroshio. On one hand, stronger updraft over the land maintains large droplets higher in the upper part of the cloud layer. On the other hand, drizzle and rain appear to form lower down in the oceanic cloud layers due to the weaker updraft (Nakajima et al., 2010). We

consider that clouds over the Kuroshio grow in the upper part of the cloud layer due to the strong updraft, such as land clouds. The cloud particles grew in the precipitating clouds at the center part from the top of the cloud through the collisional growth and gradually fell due to gravity. The growth process of cloud particles occurred from the central to lower part of the cloud. However, MaxZe would be reached at the center of the cloud by with an attenuation of radar reflectivity by the existence of the large particles in the lower part of the cloud.

## 4. Summary and Conclusions

This study clarified the effects of SST changes in the Kuroshio Current on the vertical structure of clouds using CloudSat products and meteorological data. Surface wind convergence, strong updrafts from the lower to the middle troposphere, and high relative humidity occurred over the Kuroshio for the target period, and we observed that the area with high water

content and radiative heating values increased with increasing (decreasing) skin-temperature (LTS). We showed that the strength of precipitation inside the cloud increased around the middle layers (about 6 km) compared to that in the upper layer. The influence of the Kuroshio on precipitation intensity extended to the middle atmosphere. The drizzle clouds over ON Kuroshio have a higher chance of being geometrically thin (0.5–3 km) and thicker (7–10 km) compared to those over OFF Kuroshio. Moreover, the frequency of occurrence of the precipitating clouds with a geometric thickness of 7 to 10 km

increased over the ON Kuroshio Those clouds were geometrically thick, and the precipitating liquid water content and rainfall intensity increased inside the cloud. Strong updraft over the Kuroshio could maintain large droplets higher in clouds in the upper layer, and the MaxZe position in non-precipitating and drizzle clouds over the Kuroshio was higher than that around the Kuroshio.

These results showed that the vertical structures (rainfall intensity, cloud geometrical thickness, and MaxZe position) of

clouds were distinctly different between over the Kuroshio and around the Kuroshio. We consider that the difference in the



MaxZe position may help to better understand cloud-to-precipitation transitional process over the Kuroshio. A correct understanding of the transition process from cloud to precipitation can contribute to reducing bias with observations in the GCMs (e.g., Michibata and Takemura, 2015; Michibata et al., 2016). Because we focused on only local phenomena over the Kuroshio in this study, there may have some limitations for extending to global applications. Recently, the Non-hydrostatic

Icosahedral Atmospheric Model (NICAM) (e.g., Tomita and Satoh, 2004; Satoh et al., 2008) has made it possible to reproduce cloud-related phenomena with a horizontal resolution of less than 1 km (Miyamoto et al., 2013). Furthermore, a hybrid microphysical cloud model with a two-moment bin method (Kuba and Fujiyoshi, 2006; Kuba and Murakami, 2010) has been developed, therefore, the understanding of the smaller-scale phenomena is crucially necessary. Incorporating the results of this study into these models would greatly improve the accuracy of model simulations through motivating better

representations of the rainfall parameterization and physically-based process description. Quantitative assessment of the influence of local SST gradient on cloud and atmospheric conditions is now a fundamentally important question in addition to its detection as done in this study.

Consequently, we concluded that the Kuroshio influences not only the dynamical processes of the lower layer of the atmosphere but also the properties inside clouds. The Earth Clouds, Aerosols, and Radiation Explorer (EarthCARE) satellite

is scheduled for launch in 2019. This satellite will carry a new CPR that measures the upward and downward flow velocities inside a cloud while observing its vertical structure (Illingworth et al., 2014). EarthCARE will be able to detect thinner clouds, and its Doppler capability will provide useful information on convection, precipitating ice particles, and rainfall speeds. The cloud physical properties on the Kuroshio can probably be clarified in more detail with the data from EarthCARE.

*Acknowledgements*

The CloudSat data products of 2B-GEOPROF, 2C-RAIN-PROFILE, and ECMWF-AUX were provided by the CloudSat Data Processing Center at the Cooperative Institute for Research in the Atmosphere, Colorado State University.



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

5 Figure Captions:

**Figure 1. (a)** Mean wind convergence (divergence) at 1000 hPa (color; $10^{-5}$ $s^{-1}$) and wind vector (vector; $ms^{-1}$), (b) upward pressure velocity at 400 hPa: -ω ($10^{-2}$ Pa $s^{-1}$), (c) relative humidity at 400 hPa (%) for the target period from the Japan Meteorological Agency's Mesoscale model (MSM). (d) Mean skin-temperature (deg. C), (e) lower-tropospheric stability (K), and (f) cloud fraction for the target period using CloudSat products.

**Figure 2.** The meridional vertical sections in the thick dotted square (25–34°N, 126.5–131°E) of Figure 1(a), (a) Mean wind convergence (divergence) at 1000 hPa ($10^{-5}$ $s^{-1}$), (b) upward pressure velocity at 400 hPa: -ω ($10^{-2}$ Pa $s^{-1}$), (c) relative humidity at 400 hPa (%) for the target period from the Japan Meteorological Agency's Mesoscale model (MSM). (d) mean cloud fraction, (e) cloud water content ($gm^{-3}$), (f) precipitating liquid water content ($gm^{-3}$), (g) precipitating ice water content: $gm^{-3}$, (h) total water content ($gm^{-3}$), (i) shortwave cloud radiative heating rate (Kday-1, upper panel) and skin temperature (deg. C, lower panel), (j) longwave cloud radiative heating rate (Kday-1, upper panel) and lower-tropospheric stability (K, lower panel), (k) net cloud radiative heating rate (Kday-1) using CloudSat products.

**Figure 3.** Two-dimensional Probability Density Function (PDF) for (a) ON Kuroshio, (b) OFF Kuroshio, and (c) ON–OFF
20 Kuroshio, and Contoured Frequency by Altitude Diagrams (CFADs) for (d) ON Kuroshio, (e) OFF Kuroshio, and (f) ON Kuroshio–OFF Kuroshio.



**Figure 4.** Difference in the frequency of the geometrical thickness (ON Kuroshio-OFF Kuroshio) according to three

categories of MaxZe for (a) non-precipitating, (b) drizzle, and (c) precipitating.

**Figure 5.** The relationship between the fraction of maximum radar reflectivity (MaxZe) position and altitude according to

5   the three categories of MaxZe for (a) non-precipitating, (b) drizzle, and (c) precipitating. The fraction of the MaxZe position

for the scaled cloud layer according to the three categories of MaxZe for (a) non-precipitating, (b) drizzle, and (c)

precipitating clouds.





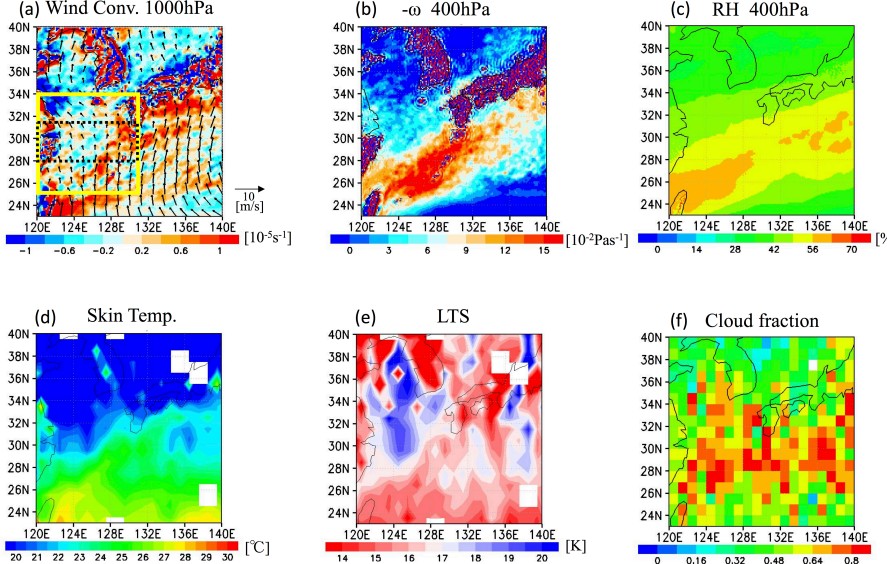

**Figure 1. (a)** Mean wind convergence (divergence) at 1000 hPa (color; $10^{-5}$ s$^{-1}$) and wind vector (vector; ms$^{-1}$), (b) upward pressure velocity at 400 hPa: -ω ($10^{-2}$ Pa s$^{-1}$), (c) relative humidity at 400 hPa (%) for the target period from the Japan Meteorological Agency's Mesoscale model (MSM). (d) Mean skin-temperature (deg. C), (e) lower-tropospheric stability (K), and (f) cloud fraction for the target period using CloudSat products.





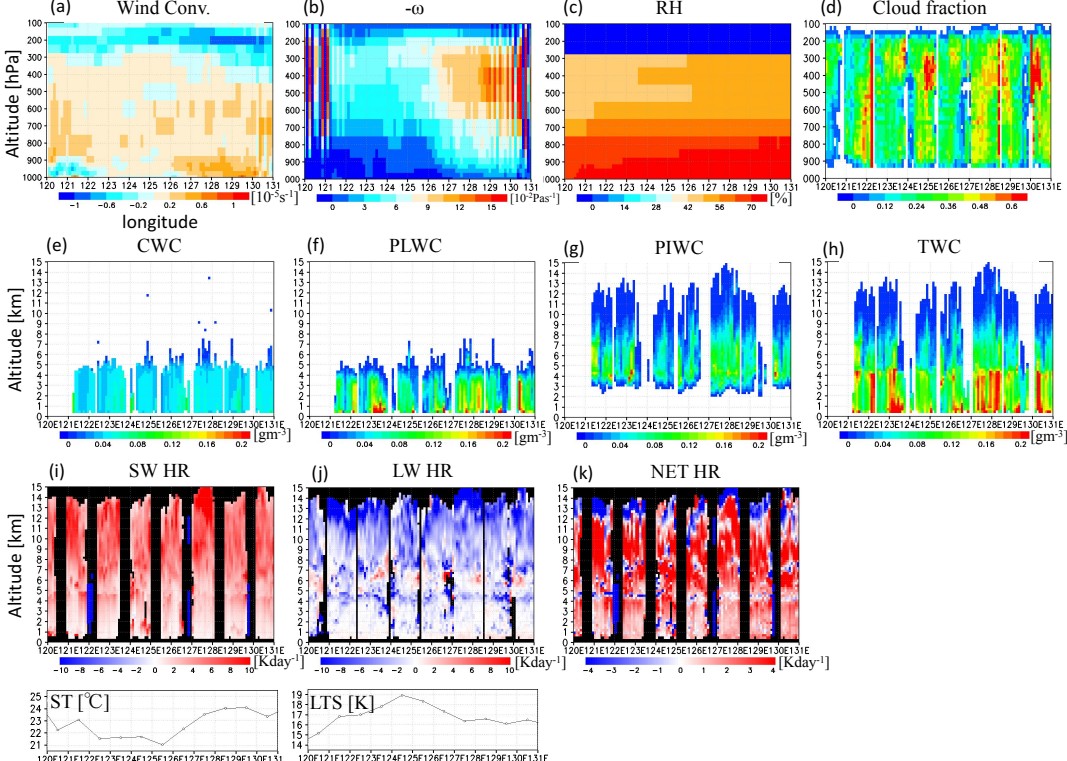

**Figure 2.** The meridional vertical sections in the thick dotted square (25–34°N, 126.5–131°E) of Figure 1(a), (a) Mean wind convergence (divergence) at 1000 hPa ($10^{-5}$ s$^{-1}$), (b) upward pressure velocity at 400 hPa: -ω ($10^{-2}$ Pa s$^{-1}$), (c) relative humidity at 400 hPa (%) for the target period from the Japan Meteorological Agency's Mesoscale model (MSM). (d) mean cloud fraction, (e) cloud water content (gm$^{-3}$), (f) precipitating liquid water content (gm$^{-3}$), (g) precipitating ice water content: gm$^{-3}$, (h) total water content (gm$^{-3}$), (i) shortwave cloud radiative heating rate (Kday$^{-1}$, upper panel) and skin temperature (deg. C, lower panel), (j) longwave cloud radiative heating rate (Kday$^{-1}$, upper panel) and lower-tropospheric stability (K, lower panel), (k) net cloud radiative heating rate (Kday$^{-1}$) using CloudSat products.





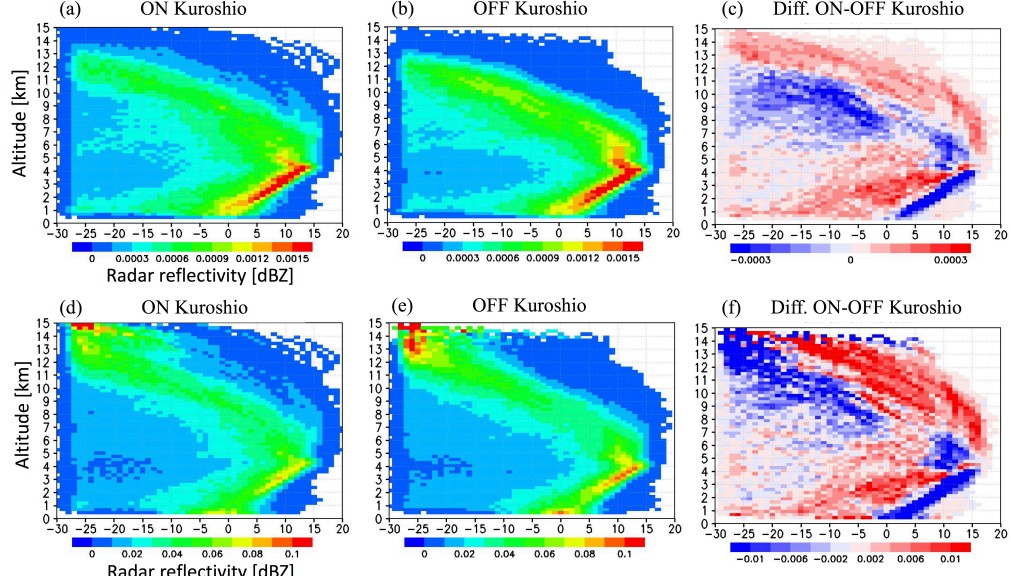

**Figure 3.** Two-dimensional Probability Density Function (PDF) for (a) ON Kuroshio, (b) OFF Kuroshio, and (c) ON–OFF Kuroshio, and Contoured Frequency by Altitude Diagrams (CFADs) for (d) ON Kuroshio, (e) OFF Kuroshio, and (f) ON Kuroshio–OFF Kuroshio.





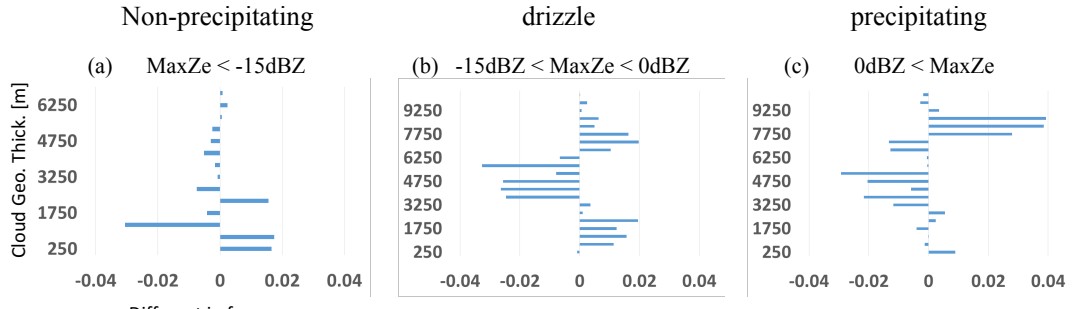

5 **Figure 4.** Difference in the frequency of the geometrical thickness (ON Kuroshio-OFF Kuroshio) according to three

categories of MaxZe for (a) non-precipitating, (b) drizzle, and (c) precipitating.



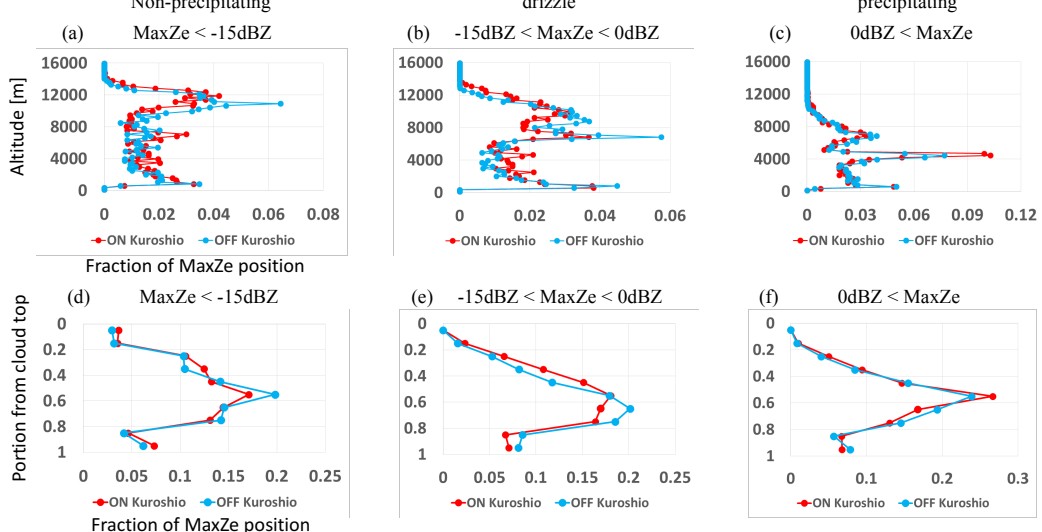

**Figure 5.** The relationship between the fraction of maximum radar reflectivity (MaxZe) position and altitude according to the three categories of MaxZe for (a) non-precipitating, (b) drizzle, and (c) precipitating. The fraction of the MaxZe position for the scaled cloud layer according to the three categories of MaxZe for (a) non-precipitating, (b) drizzle, and (c) precipitating clouds.