# Peer review of "Assessing the impact of the Kuroshio Current on vertical cloud structure using CloudSat data"

_Atmospheric Chemistry and Physics, 2017_

## Referee Comment (RC1) · Anonymous Referee #1 · 3 Mar 2018

Review of 'Assessing the impact of the Kuroshio Current on vertical cloud structure using CloudSat data' by A. Yamauchi et al.

This manuscript investigates the response of clouds to the Kuroshio current in the East China Sea region. CloudSat products are used to provide cloud properties and precipitation profiles, combined with regional re-analyses in order to provide a meteorological context to the study. Analyses are here limited to summer months (15 May - 15 June) from 2007 to 2010. The authors' main conclusions are that the Kuroshio current leads to stronger precipitation in thick (convective) clouds, which also become more frequent.

An important motivation for this study is the need to provide regional observational constrains to climate or weather prediction models. The authors point in particular to limitations in auto-conversion rate parameterizations in GCMs or the need to thoroughly evaluate the representation of cloud and precipitation structure in new high-resolution cloud resolving models (CRMs) such as NICAM. I fully agree with the importance of these questions, but find that using 3 years of satellite data makes the results presented here fall in between of these two objectives. A longer time period is needed to establish more rigorous climatological conclusions useful to GCMs, whereas more precise case study analyses would be needed to evaluate CRMs. I would advise for the former solution, as discussed below.

I also fail to be fully convinced by some of the conclusions made here, as the presented cloud properties observed in ON and OFF Kuroshio regions (terms used in the manuscript to identify areas under the influence of the current or not) do not always clearly seem significantly different. The conclusions are not illogical with expectations, but I would advise the authors to perhaps change their definitions of the ON and OFF regions in order to make the differences more obvious (see comments below). More in-depth analyses and a stronger story line that links the figures together could also help making some of the arguments more convincing. This could be done by introducing a short section that discusses the overall conclusions based on all figures.

Nevertheless, I think that this study is of interest and can significantly be improved by adding more data and clarifying a few points. It is generally well written and, providing more in-depth analyses, could fit within the scope of ACP. I therefore advise for publication after major revision. Further details are provided below.

Major comments:

1. p2, l1-2: The authors cite two papers by Koike et al. (2012, 2016) that investigate the impact of aerosol on liquid clouds in relation to Kuroshio current. It would be nice to mention this effect and clarify how the present study fits in the context of this previous literature. Could some of the cloud and precipitation changes observed in this manuscript result from fast cloud adjustments to aerosol perturbations induced by the current? I realize that this is not the main topic here, and disentangling aerosol-cloud-meteorology effects is extremely difficult, but this issue could be briefly addressed.

2. p3, l17-18: I strongly encourage the authors to expand their analyses to the entire A-Train period (2006-2016). This would greatly improve the statistical significance of the results presented here, and I think could even help the authors to make their conclusions stronger by reducing weather noise. The data is freely available and I do not see any reason to only use 3 years of data. Perhaps because the night-time data is not available later on? Also, have you have merged day and night overpasses in the analyses and, if so, are there any consequences on the results by comparison to using day-only and night-only statistics?

3. p3, l20-23: Using a strict SST threshold is perhaps not the best option to determine the ON and OFF regions. This method implies that transition areas are included and could blur the expected changes in statistics of cloud properties and precipitation between the two regions. Another option could be to select ON and OFF regions based on separate ranges of temperatures, for instance corresponding to the first and last quartiles (or other percentiles, depending on the desired sensibility) of region-mean SSTs. This way, both regimes are better defined and can be distinguished.

4. Section 3: As previously mentioned, and in relation to the previous point, the differences between properties observed in ON and OFF regions aren't always obvious. For example, the authors have mixed interpretations of Fig. 2, sometimes stating that no clear differences are observed (e.g. p5 l2-3, l6-7, and I'd agree with that) and later that the Kuroshio impacts heating at high altitude. It is not clear to me from Fig. 2k where this impact is, could you clarify? Also, based on Figure 5 (analyses in section 3.4) I do not see any clear signal of perturbation by the Kuroshio current, except perhaps in 4e. More statistics (more A-Train data) could help clarify if this is within weather noise or not. In general, all the 3.x subsections appear a bit as lists of figure descriptions without a consistent in-depth analysis, until the very end where interesting arguments based on cloud processes are provided (beginning of p.8). It would be good to improve how each figure and their respective results fit all together, to make the final conclusions more convincing. I had some troubles understanding towards the end how all the presented results connect, maybe an extra section would help.

Minor comments:

1. p2, l.11: This is a detail, but isn't the across-track resolution of 1.4 km? This information, and the following technical details, could go in section 2.

2. p4, l.18: Any reason to subset the previous region (yellow box)? Especially that this thick dotted rectangle region only is used in section 3.1.

3. p2, l.12: Analysis of Fig. 2i,j: What are the vertical black bands?

4. p2, l.17: net radiative heating here means SW + LW?

5. p5, l.24: Could you clarify what are "contoured frequencies by altitude diagrams" and their advantage by comparison to the previous PDF?

6. p6, l.8: "for higher altitudes": Do you mean above 6 km? The statistical significance doesn't seem high in regions where the difference if positive at higher altitude. It seems more clear from this figure that heavy precipitation below 6-km is reduced but there is more drizzle.

7. p6, l.15: Can you precise how the geometrical thickness is computed? Please keep in mind that CloudSat is sensitive to the surface echo (so the cloud base of low clouds is difficult to get) and not sensitive to thin layers (so could miss cloud-top). I am not sure how this impacts the conclusions made here.

8. p7, l2-3: "the frequency of occurrence of the precipitating clouds with a geometric thickness of 7 to 10 km significantly increased ON Kuroshio." - True, but does it mean that thick clouds precipitate more or that there are more thick clouds in ON Kuroshio regions (consequence of stronger updrafts). This is an example where coupled analyses with results from previous figures would be helpful.

9. Figure 4: How did you bin the geometric thickness? How is the PDF normalized?

Technical corrections:

1. p2, l.4: "General" instead of "Generation" or "Generation of general"? I assume that the authors refer to the "too bright too few" problem, but the sentence is a bit confusing.

2. p2, l.12: "cloud particle size, which ranges from drizzle to precipitation" can be misleading. Replace by "large cloud particles and hydrometeors"?

3. p4, l.10: why "(divergence)"?

4. p4, l.19: Are the sections not meridional instead of zonal?

5. p6, l.4: "for both" instead of "both"?

---

## Referee Comment (RC2) · Anonymous Referee #2 · 14 Mar 2018

Atmospheric Chemistry and Physics Discussion: "Assessing the impact of the Kuroshio Current on vertical cloud structure using CloudSat data" Authors: A. Yamauchi, K. Kawamoto, A. Manda, and J. Li Manuscript Number: acp-2017-1134

This study uses four total months (May 15 – June 15 for 2007-2010) of CloudSat radar reflectivity data, rain profile data, and heating rates data to investigate the cloud and precipitation vertical structure near and adjacent to the Kuroshio Current, including the impact of the local warm waters and strong SST gradients during this late spring/early summer period. Complementary, collocated ECMWF-AUX reanalysis data are used for altitude and temperature profiles, and large-scale circulation data at a fine resolution, including profiles of vertical velocity, divergence, and winds come from the Japan Meteorological Agency's Mesoscale Model (MSM). The justification for this time pe-

riod is the strong SST gradients over the East China Sea (ECS), specifically the SST difference between the Kuroshio and non-Kuroshio areas, for which an SST threshold of 24°C separates the two regimes ("ON-Kuroshio" and "OFF-Kuroshio"). Maps of large-scale meteorological/climate variables are first presented, including dynamic and thermodynamic quantities of interest in and around the domain chosen, and then vertical profiles along a cross section of the domain are collected and shown. The presentation of the remainder of the study predominantly includes comparing cloud and dBZ profiles for the ON-Kuroshio and OFF-Kuroshio in a variety of ways, including PDFs of dBZ profiles, and then examining PDFs of different cloud types, including non-precipitating, drizzle, and precipitating regimes. The main take-away message is that rain intensity in the mid-troposphere is stronger over the defined Kuroshio regime versus surrounding areas, with a greater frequency of precipitation from geometrically thick clouds over ON-Kuroshio. Even in drizzling and non-precipitating clouds, the authors show a slight increase of the altitude of dBZmax in ON-Kuroshio profiles. All-in-all, despite the effort by the authors to separate the Kuroshio-influenced atmosphere from the adjacent areas in documenting possible vertical cloud structure differences using active radar and some auxiliary data, the limited amount of data analyzed, only four total months, is unfortunately a significant shortcoming of this study. CloudSat already suffers from sampling rather sparse data, due to the inherent thin curtain nature of its sampling, and collecting only one month per year of data arguably does not provide a significant-enough sample size for which to draw more robust conclusions. Indeed, the data striping in Figure 2 of profiles of cloud fraction, cloud water content, precipitating liquid/ice water content, and total water content, as well as the structures of longwave and shortwave heating rates, underscore that some areas of the cross section do not even get sampled out of the total of four months assessed. This is indeed problematic, and it's difficult to draw any meaningful conclusions from any of the CloudSat-borne quantities in that figure, save perhaps for the discrimination of net heating versus cooling rate profiles (panel k). The authors rationalize the one-month per year analysis because of the maximum strength of SST gradients, but how about at least doubling

that (April 15 – July 15), and then possibly adding more years as well? The goal needs to be at least the minimum of profiles that would provide full coverage and remove the striping for the profile analysis in Fig. 2, which at this time, except for the large-scale variables (e.g. convergence, vertical velocity, and RH), has little significance for the study. It may be helpful to bring in other A-Train datasets, such as Aqua MODIS data, to provide additional information about cloud fraction, cloud optical depth, vertical structure, and perhaps even effective radius. The latter could be quite beneficial in particular to help quantify the relationships between the radar reflectivity and radius between ON-Kuroshio and OFF-Kuroshio, particularly as they may relate to the results shown in Figs. 3 – 5. Furthermore, given that CloudSat becomes attenuated at ∼15 dBZ, it may be useful to include a sensor which provides precipitation for more heavily raining clouds, such as AMSR-E, to pin down the differences particularly for some of the deeper convection periods captured in this study. Bringing in AMSR-E would allow a more quantitative analysis of the contribution of different cloud heights to total precipitation between ON-Kuroshio and OFF-Kuroshio. A useful exercise may be compositing against altitude of dBZ_max in both regimes; if altitude is normalized then the explicit role of surface conditions and large-scale vertical velocity in Kuroshio versus OFF-Kuroshio could be analyzed. Similarly, examining the vertical structure against different rain rate categories in a more holistic way may be more satisfying than the one-category only now ("precipitating" category), and would provide potential physical insights as well as useful information for climate model parameterizations. The authors should also consider performing analyses of cloud vertical structure, vertical velocity, and some other pertinent cloud properties already shown as a function of SST. There may be no need to partition Kuroshio versus adjacent areas, as SST itself may naturally distill the results. Finally, the manuscript needs to be proofread by a professional English editor, as the tenses (e.g. past and present) jump around improperly; indeed most of the study should be in the present tense, but much of it is in past tense. A few explicit examples of this are provided at the very end of this review, as well as a non-exhaustive list of grammatical errors and typos. Overall, this paper may eventually

be publishable, but it will require extensive and major revisions, as well as additional data, for it to be a sufficiently complete study.

Specific Comments: 1. Figure 1: The yellow box, representing the target region, as well as the sub-domain represented by the thick-dashed box, should be shown in each panel of Figure 1, which would aid the reader in orientating the main features more readily from each of the fields displayed. Again, adding considerably more data, and possibly MODIS cloud fraction, would make Fig. 1f much more meaningful than it is now, which stands currently as a fairly chaotic field of cloud fraction due to the noisiness. Please also consider an improved color scheme, especially for Fig. 1d, which shows the skin temperature. The gradations are very subtle between about 23°-26°C, even though this encompasses the critical threshold for defining ON-Kuroshio and OFF-Kuroshio. 2) Figure 2: As stated in the summary and overarching comments at the beginning of this review, many of the CloudSat-derived or retrieved profiles are almost meaningless here, partly because of the sparse and limited sampling (with data striping!), and perhaps in some cases, because of the color schemes chosen. For the SW HR, LW HR, and Net HR plots, while it is possible to distinguish between reds (warming) and blues (cooling), it is very difficult to discern the seemingly more subtle differences across the cross section analyzed. Also, there appears to be an inconsistency between the manuscript text and the caption in Figure 2 – the latter states the thick dotted box between (25-34N, 126.5-131E), but the figures themselves show longitude values between 120 – 131E, as does the box itself in Fig. 1. Also, the latitude range from Fig. 1 is 28-31.5N, which is also stated in the text body, but this is different from the caption of Fig. 2. Please correct. 3) Line 3, page 4: Please consider adding "Frisch et al. 1995" for an additional, more historical citation – e.g. this is an early paper which uses -15 dBZ to discriminate between non-drizzling and drizzling/precipitating clouds. Reference: Frisch, A. S., C. W. Fairall, and J. B. Snider, 1995: Measurement of stratus cloud and drizzle parameters in ASTEX with a Ka-band Dopper radar and a microwave radiometer. J. Atmos. Sci., 52, 2788-2799. 4) Line 8, page 5: The sentence: "The total water content (TWC: CWC+PLWC+PIWC) corresponded to LTS,

which peaked around 124.5E", is very vague and confusing, and needs to be rewritten. 5) Line 18, page 5: The phrase, "As previously described, TWC increased over the Kuroshio" is rather difficult for me to discern from Fig. 2k. Perhaps the black striping and the color scheme make this result a difficult one to view. In another vein, if the authors decide to assess any other A-Train data, examining cloud radiative forcing in a similar way from CERES, including Longwave, Shortwave, and net, might be complementary to Figure 2 and the paper in general. The real question is – do clouds over the Kuroshio have a larger net TOA radiative effect? Profiles of cloud radiative effects can also be assessed from CERES, if there is space to perform this analysis. 6) Lines 4-5, page 6: Please consider re-writing the sentence as follows: "These results show that clouds with the highest rainfall intensity measurable by CloudSat at lower altitudes (1-6 km) are common in the target region." 7) Lines 4-7, page 7: Why are mid-thickness drizzling clouds more abundant in the OFF-Kuroshio region (Fig. 4b)? Is this because the ascending motion is weaker and more bottom-heavy than over the Kuroshio current, leading to a greater abundance presumably of mid-level clouds? This is also the case for precipitating clouds; mid-thickness clouds are more pervasive in the OFF-Kuroshio regions. Can we say anything about total precipitation from this Figure (or Figure 5)? It would be interesting to know how much the different categories contribute to total precipitation, and this is where an independent, additional sensor which does not attenuate for dBZ>15 dBZ would be helpful, such as AMSR-E. 8) Lines 25-26, page 7: "...taking 0 and 1 at the cloud top and the cloud base, respectively..." There's no need to repeat this here, as it is already explicitly described at the beginning of that paragraph.

Grammatical Suggestions and an Incomplete List of Typos (Please have a professional English editor carefully proof this manuscript) 1) As an illustration of the tense problem reported above, Lines 6-10, page 3 are in past tense, but this is inappropriate as it discusses the organizational structure of the paper – e.g. should instead be: "In section 2, we show the data . . . In section 3.1, we describe the influence. . .". The authors seesaw between past and present tense, sometimes opening paragraphs in present tense, but

then reverting to past tense by mid-paragraph. Please correct this – it happens during so many instances that it's not convenient to enumerate them all here. 2) Line 1, page 5: change "less than" to "west of" 3) Line 4, page 6: add "over" before "both" 4) Line 20, page 8: a period is missing after "ON Kuroshio". 5) Line 21, page 8: change "updraft" to "updrafts"

---

## Author Comment (AC1) · 25 Apr 2018

Dear editor and reviewers,

Please find attached a revised version of our manuscript "Assessing the impact of the Kuroshio Current on vertical cloud structure using CloudSat data (acp-2017-1134)", which we would like to resubmit for publication as Atmospheric Chemistry and Physics.

Comments of you and reviewers were highly insightful and enabled us to greatly improve the quality of our manuscript. The following pages are our point-by-point responses to each of the comments of the reviewers.

Revisions in the text are shown using yellow highlight for additions. We hope that the revisions in the manuscript and our accompanying responses will be sufficient to make our manuscript suitable for publication in Atmospheric Chemistry and Physics.

We shall look forward to hearing from you at your earliest convenience.

Yours sincerely,

Akira Yamauchi
Atmosphere and Ocean Research Institute, The University of Tokyo
E-mail: yamauchi@aori.u-tokyo.ac.jp

RC: Referee comment

AC: Author comment

**Response to Referee #1**

Major comments:

RC: p2, l1-2: The authors cite two papers by Koike et al. (2012, 2016) that investigate the impact of aerosol on liquid clouds in relation to Kuroshio current. It would be nice to mention this effect and clarify how the present study fits in the context of this previous literature. Could some of the cloud and precipitation changes observed in this manuscript result from fast cloud adjustments to aerosol perturbations induced by the current? I realize that this is not the main topic here, and disentangling aerosol-cloud-meteorology effects is extremely difficult, but this issue could be briefly addressed.

AC: Fast cloud adjustments to aerosol perturbations could be induced by the Kuroshio, and then some of the cloud and precipitation changes occurred. However, it is difficult to investigate the effect of aerosols with CloudSat only. For future work, we will investigate the effect of aerosol using CALIPSO data and climate model outputs.

   We added the text in section 4 as follows. 'Strong updrafts over the Kuroshio affected the vertical structure and microphysics of low altitude non-precipitating water clouds (Koike et al., 2012, 2016). Fast cloud adjustments to aerosol perturbations could be induced by the Kuroshio, and then some of the cloud and precipitation changes occurred. However, it is difficult to investigate the effect of aerosols with CloudSat only. For future work, we will investigate the effect of aerosol using CALIPSO data and climate model outputs.' (P9, L24-P10, L3)

RC: p3, l17-18: I strongly encourage the authors to expand their analyses to the entire A-Train period (2006-2016). This would greatly improve the statistical significance of the results presented here, and I think could even help the authors to make their conclusions stronger by reducing weather noise. The data is freely available and I do not

see any reason to only use 3 years of data. Perhaps because the night-time data is not available later on? Also, have you have merged day and night overpasses in the analyses and, if so, are there any consequences on the results by comparison to using day-only and night-only statistics?

AC: We expanded the target period to 2016. Although we analyzed satellite data separately between day and night, there was no large difference in the results.

RC: p3, l20-23: Using a strict SST threshold is perhaps not the best option to determine the ON and OFF regions. This method implies that transition areas are included and could blur the expected changes in statistics of cloud properties and precipitation between the two regions. Another option could be to select ON and OFF regions based on separate ranges of temperatures, for instance corresponding to the first and last quartiles (or other percentiles, depending on the desired sensibility) of region-mean SSTs. This way, both regimes are better defined and can be distinguished.

AC: We performed reanalysis using different thresholds in May ($23.08^\circ$C) and June ($24.70$°C). The thresholds were higher by 1 $^\circ$C than the regional average temperatures in each month. (P3, L24 - P4, L2)

RC: Section 3: As previously mentioned, and in relation to the previous point, the differences between properties observed in ON and OFF regions aren't always obvious. For example, the authors have mixed interpretations of Fig. 2, sometimes stating that no clear differences are observed (e.g. p5 l2-3, l6-7, and I'd agree with that) and later that the Kuroshio impacts heating at high altitude. It is not clear to me from Fig. 2k where this impact is, could you clarify? Also, based on Figure 5 (analyses in section 3.4) I do not see any clear signal of perturbation by the Kuroshio current, except perhaps in 4e. More statistics (more A-Train data) could help clarify if this is within weather noise or not. In general, all the 3.x subsections appear a bit as lists of figure descriptions without a consistent in-depth analysis, until the very end where interesting arguments based on cloud processes are provided (beginning of

p.8). It would be good to improve how each figure and their respective results fit all together, to make the final conclusions more convincing. I had some troubles understanding towards the end how all the presented results connect, maybe an extra section would help.

AC: We added figures of cloud properties and radiation at TOA and surface (Fig. 2, 3) using some satellite data. We could show the characteristics more clearly over the Kuroshio. (P4, L22-P5, L15)

Minor comments:

RC: p2, l.12: This is a detail, but isn't the across-track resolution of 1.4 km? This information, and the following technical details, could go in section 2.

AC: We replaced '1.3 km' to '1.4 km', and this information moved to section 2. (P3, L10)

RC: p4, l.18: Any reason to subset the previous region (yellow box)? Especially that this thick dotted rectangle region only is used in section 3.1.

AC: The yellow box indicated an area with a large difference between east and west region in atmospheric condition on the map.

RC: p2, l.12: Analysis of Fig. 2i,j: What are the vertical black bands?

AC: The blacks bands mean missing values.

RC: p2, l.17: net radiative heating here means SW + LW?

AC: Net radiative heating means SW + LW, and We added 'SW + LW' in the figure caption.

RC: p5, l.24: Could you clarify what are "contoured frequencies by altitude diagrams" and their advantage by comparison to the previous PDF?

AC: Contoured Frequency by Altitude Diagrams (CFADs) is defined as the probability distribution of radar reflectivity normalized at each altitude. CFADs can get information on precipitation intensity for each altitude. We added those descriptions in section 3.2.

RC: p6, l.8: "for higher altitudes": Do you mean above 6 km? The statistical significance doesn't seem high in regions where the difference if positive at higher altitude. It seems more clear from this figure that heavy precipitation below 6km is reduced but there is more drizzle.

AC: 'at high altitudes' mean above 6 km. We remove a text after 'at high altitude ~', and added the text as follows. 'heavy precipitation below 6km is reduced but there is more drizzle over the Kuroshio.' (P7, L7)

RC: p6, l.15: Can you precise how the geometrical thickness is computed? Please keep in mind that CloudSat is sensitive to the surface echo (so the cloud base of low clouds is difficult to get) and not sensitive to thin layers (so could miss cloud-top). I am not sure how this impacts the conclusions made here.

AC: We used CloudSat product only. You are right, we cannot observe thin cloud top and cloud bottom near the ground surface. Even if we use the radar-lidar product, this analysis cannot be performed because data on the radar reflectivity cannot be obtained in the thin clouds and the near the ground surface.

RC: p7, l2-3: "the frequency of occurrence of the precipitating clouds with a geometric thickness of 7 to 10 km significantly increased ON Kuroshio." - True, but does it mean that thick clouds precipitate more or that there are more thick clouds in ON Kuroshio regions (consequence of stronger updrafts). This is an example where coupled analyses with results from previous figures would be helpful.

AC: It means that there are more thick clouds over the Kuroshio regions. We added a description of 'Geometrically thick clouds occurred more because strong updrafts (Fig 2a) took place over the Kuroshio.' (P8, L4)

RC: Figure 4: How did you bin the geometric thickness? How is the PDF

normalized?

AC: First, we calculated PDFs of geometrical thickness in each 500m for ON and OFF Kuroshio clouds, and then we took the difference of two PDFs, respectively.

Technical corrections:

RC: p2, l.4: "General" instead of "Generation" or "Generation of general"? I assume that the authors refer to the "too bright too few" problem, but the sentence is a bit confusing.

AC: We replaced 'generation' with 'general' and added description of 'so-called 'too few, too bright' problem.' to the end of the sentence. (P2, L3-5)

RC: p2, l.12: "cloud particle size, which ranges from drizzle to precipitation" can be misleading. Replace by "large cloud particles and hydrometeors"?

AC: We replaced 'cloud particle size, which ranges from drizzle to precipitation' with 'large cloud particles and hydrometeors'. (P2, L11-12)

RC: p4, l.10: why "(divergence)"?

AC: we deleted "(divergence)".

RC: p4, l.19: Are the sections not meridional instead of zonal?

AC: We replaced 'zonal' with 'meridional' (P5, L17)

RC: p6, l.4: "for both" instead of "both"?

AC: We added 'for'. (P7, L3)

**Response to Referee #2**

Specific Comments:

RC: 1. Figure 1: The yellow box, representing the target region, as well as the sub-domain represented by the thick-dashed box, should be shown in each panel of Figure 1, which would aid the reader in orientating the main features more readily from each of the fields displayed. Again, adding considerably more data, and possibly MODIS cloud fraction, would make Fig. 1f much more meaningful than it is now, which stands currently as a fairly chaotic field of cloud fraction due to the noisiness.

AC: We added yellow box and dashed box in each panel of Fig.1. We added MODIS cloud fraction (Fig. 1g), and we show that cloud fraction of CloudSat could have a possibility of underestimation compared to that of MODIS. (P4, L23 -25)

RC: Please also consider an improved color scheme, especially for Fig. 1d, which shows the skin temperature. The gradations are very subtle between about 23∘ - 26∘C, even though this encompasses the critical threshold for defining ON-Kuroshio and OFF-Kuroshio. 2)

AC: We improved color scheme in Fig. 1d.

RC: Figure 2: As stated in the summary and overarching comments at the beginning of this review, many of the CloudSat-derived or retrieved profiles are almost meaningless here, partly because of the sparse and limited sampling (with data striping!), and perhaps in some cases, because of the color schemes chosen. For the SW HR, LW HR, and Net HR plots, while it is possible to distinguish between reds (warming) and blues (cooling), it is very difficult to discern the seemingly more subtle differences across the cross section analyzed. Also, there appears to be an inconsistency between the manuscript text and the caption in Figure 2 – the latter states the thick dotted box between (25-34N, 126.5-131E), but the figures themselves show longitude values between 120 – 131E, as does the box itself in Fig. 1. Also, the latitude range from Fig. 1 is 28-31.5N, which is also stated in the text body, but this is different from the caption of Fig. 2. Please correct. 3)

AC: We expanded the target period to 2016, and sample size increased. We corrected the region (28–31.5°N, 120–131°E) of the manuscript and the caption in Fig.2.

RCs: Line 3, page 4: Please consider adding "Frisch et al. 1995" for an additional, more historical citation – e.g. this is an early paper which uses -15 dBZ to discriminate between non-drizzling and drizzling/precipitating clouds. Reference: Frisch, A. S., C. W. Fairall, and J. B. Snider, 1995: Measurement of stratus cloud and drizzle parameters in ASTEX with a Ka-band Dopper radar and a microwave radiometer. J. Atmos. Sci., 52, 2788-2799. 4)

AC: We added 'Frisch et al. 1995' as one of the references. (P4, L8-9)

RC: Line 8, page 5: The sentence: "The total water content (TWC: CWC+PLWC+PIWC) corresponded to LTS, which peaked around 124.5E", is very vague and confusing, and needs to be rewritten. 5)

AC: We added a description of 'LTS peaked around 124.5E°, LTS peaked around 124.5E°, and TWC was the lowest there.' (P6, L6)

RC: Line 18, page 5: The phrase, "As previously described, TWC increased over the Kuroshio" is rather difficult for me to discern from Fig. 2k. Perhaps the black striping and the color scheme make this result a difficult one to view. In another vein, if the authors decide to assess any other A-Train data, examining cloud radiative forcing in a similar way from CERES, including Longwave, Shortwave, and net, might be complementary to Figure 2 and the paper in general. The real question is – do clouds over the Kuroshio have a larger net TOA radiative effect? Profiles of cloud radiative effects can also be assessed from CERES, if there is space to perform this analysis. 6)

AC: We reduced black striping from Fig. 2k, and we added figures of TOA and surface flux (Figure 3 a to f) from CERES products. Those figures showed that the cooling effect worked at the surface over the Kuroshio in the same manner as TOA. (P5, L8-15)

RC: Lines 4-5, page 6: Please consider re-writing the sentence as follows: "These results show that clouds with the highest rainfall intensity measurable by CloudSat at lower altitudes (1-6 km) are common in the target region." 7)

AC: We rewrote the sentence as follows 'These results show that clouds with the highest rainfall

intensity measurable by CloudSat at lower altitudes (1-6 km) are common in the target region.'
(P7, L2-3)

RC: Lines 4-7, page 7: Why are mid-thickness drizzling clouds more abundant in the OFF-Kuroshio region (Fig. 4b)? Is this because the ascending motion is weaker and more bottom-heavy than over the Kuroshio current, leading to a greater abundance presumably of mid-level clouds? This is also the case for precipitating clouds; mid-thickness clouds are more pervasive in the OFF-Kuroshio regions. Can we say anything about total precipitation from this Figure (or Figure 5)? It would be interesting to know how much the different categories contribute to total precipitation, and this is where an independent, additional sensor which does not attenuate for dBZ>15 dBZ would be helpful, such as AMSR-E. 8)

AC: As you suggested, the updrafts is weaker and more bottom-heavy over the OFF Kuroshio than over the ON Kuroshio. It is the reason to occur mid-thickness clouds over the OFF Kuroshio. We added a figure of mean precipitation rate (Fig. 2a). This figure showed that mean precipitation rate in target period reached 10-12 mm day$^{-1}$ over the Kuroshio, and it was twice as large as that in the surrounding Kuroshio area (5-6 mm day$^{-1}$). (P5, L3-9)

In this study, we focused on clouds that could be detected with CloudSat satellite. In future work, we will try to different precipitating categories using Ku-band and Ka-band.

RC: Lines 25-26, page 7: ". . .taking 0 and 1 at the cloud top and the cloud base, respectively. . ." There's no need to repeat this here, as it is already explicitly described at the beginning of that paragraph.

AC: We deleted the text.

RC: Grammatical Suggestions and an Incomplete List of Typos (Please have a professional English editor carefully proof this manuscript) 1) As an illustration of the tense problem reported above, Lines 6-10, page 3 are in past tense, but this is inappropriate as it discusses the organizational structure of the paper – e.g. should instead be: "In section 2, we show the data . . . In section 3.1, we describe the influence. . .". The authors seesaw between past and present tense, sometimes opening paragraphs in present tense, but then reverting to past tense by mid-paragraph.

AC: We unified the tense in each section.

RC: Please correct this – it happens during so many instances that it's not convenient to enumerate them all here. 2) Line 1, page 5: change "less than" to "west of" 3) Line 4, page 6: add "over" before "both" 4) Line 20, page 8: a period is missing after "ON Kuroshio". 5) Line 21, page 8: change "updraft" to "updrafts"

AC: We rewrote these points.